# Relationship between Endotoxin Content in Vaccine Preclinical Formulations and Animal Welfare: An Extensive Study on Historical Data to Set an Informed Threshold

**DOI:** 10.3390/vaccines12070815

**Published:** 2024-07-22

**Authors:** Federica Baffetta, Raffaella Cecchi, Eva Guerrini, Simona Mangiavacchi, Gilda Sorrentino, Daniela Stranges

**Affiliations:** GSK, 53100 Siena, Italy; federica.x.baffetta@gsk.com (F.B.); raffaella.x.cecchi@gsk.com (R.C.); eva.x.guerrini@gsk.com (E.G.); simona.x.mangiavacchi@gsk.com (S.M.); gilda.x.sorrentino@gsk.com (G.S.)

**Keywords:** endotoxins, preclinical studies, in vivo, animal welfare, threshold

## Abstract

The most widely known pyrogen impurity in vaccines is the Gram-negative bacterial endotoxin lipopolysaccharide (LPS). When administered at toxic doses, endotoxin triggers inflammatory responses, which lead to endotoxic shock. The literature on endotoxic content (EC) for preclinical vaccines’ formulations used in animal studies is very poor, and the recommended thresholds are solely based on commercial vaccine limits set for humans and are, therefore, not connected to the actual impact of EC on animal welfare for species used in preclinical research studies. An extensive study to evaluate the presence of a potential relationship between endotoxin content in formulations administered to mice (the most common species used in preclinical research studies) and their welfare was conducted to calculate an EC threshold for formulations of candidate vaccines. Three years of historical data, from more than 500 formulations of different antigen types (i.e., proteins, glycoconjugates, OMV/GMMA) injected into more than 5000 mice, was evaluated with two alternative statistical methodologies, both demonstrating that there is no significant relationship between actual endotoxin levels and mouse welfare. The calculation of thresholds was, therefore, performed by consistency versus formulations that demonstrated no impact on animal welfare.

## 1. Introduction

Endotoxin impurity content is an attribute monitored in commercial vaccines.

The vaccine’s endotoxin content threshold is usually reported in their release specifications. However, there are groups of vaccines with extremely high endotoxin contents such as toxoids, and on the other side, others with very low endotoxin content, such as those containing recombinant subunit proteins and gene vectors [1,2]. So far, the recommended limits for endotoxin impurity, for preclinical research in vivo studies, have been calculated solely based on the endotoxin limits acceptable for commercial vaccines for human use [1]. However, these limits were not directly correlated to animal welfare, especially to mice welfare.

Vertebrate animal models, such as rodents, are indeed commonly used for preclinical in vivo studies in research [3,4,5,6].

Animal models, including rodents, non-human primates, pigs, and zebrafish, among others, have been widely utilized due to their genetic and physiological resemblances to humans. These models have been crucial in researching a variety of conditions, such as COVID-19, metabolic disorders like diabetes and obesity, different forms of cancer, and neurological disorders like Alzheimer’s and Parkinson’s disease. Animals have significantly contributed to the progress of biomedical research, offering valuable insights into various human and animal diseases and assisting in the creation and testing of new therapeutic methods. However, the use of animals also presents ethical issues and necessitates strict adherence to the 3Rs principles (Replacement, Reduction, and Refinement) to ensure their humane treatment. Recently, there has been an increasing interest in creating and using non-animal models, such as organoids and in silico models, which can provide human-relevant data and decrease dependence on animal testing. In general, both animal and non-animal biological models have their drawbacks, but they continue to be an essential tool in biomedical research, significantly enhancing our understanding of health and disease and the creation of new treatments [7]. Rodents, especially mice and rats, are the most frequently used species in biomedical research because of their physiological similarity to humans, their small size, ease of handling, rapid reproduction, and cost-effectiveness [3].

Mice are considered endotoxin-resistant when compared to other species, as they can tolerate extremely high doses [8,9,10,11]. Their susceptibility to endotoxin toxicity is variable, depending on the administration route, strain, and even gender [11,12]. Endotoxemia in rodents is known to affect the gastro-intestinal and respiratory systems. After endotoxin inoculation, mice may exhibit non-specific signs (i.e., general signs shared with different causes), such as anorexia, dyspnea (labored breathing), body weight loss, poor body condition, piloerection (ruffled fur), kyphosis (arched posture), and lack of movements due to acute inflammatory response [11,12]. Although the pattern of inflammation is similar in human and murine models, mice do not present inflammatory signs when treated with endotoxin doses that cause inflammation in humans [10], and the genomic responses to inflammation significantly differ between humans and mice [13,14].

It has been reported that the median lethal dose of endotoxin/LPS in mice is approximately 10–12 mg/kg [9,10]. Considering the following parameters, the minimal mouse body weight (~0.02 kg), the value of lethal dose (~10 mg/kg), and the general conversion 10 EU/mL = 1.0 ng/mL, the lethal dose in a mouse is 200,000 ng of endotoxin/mouse = 2,000,000 EU/mouse.

Animal testing of candidate vaccines is routinely performed by pharmaceutical companies to evaluate the immunogenicity of different antigens combined or not with adjuvants [15,16,17]. Many animal models specifically use mouse strains. For the statistical analysis reported in this study, experimental data from in vivo research studies performed in mice over a 3-year period in GSK Italy, in Siena, were used to obtain a wide and highly representative dataset.

Two major types of antigens were included in this study. Most of the antigens considered were purified bacterial recombinant proteins and glycoconjugates (i.e., carbohydrate-based antigens chemically bound to carrier proteins [18]), in which the purity of proteins was >85% (tested in SEC by Size Exclusion Chromatography), and the free saccharide for the glycoconjugates was below the detection limit (typically <10%). This class of antigens typically have very low endotoxin levels. The other class of antigens considered was represented by outer membrane vesicles (OMVs) [19] or generalized modules for membrane antigens (GMMAs) [20], which can be chemically extracted from whole bacteria using detergents or by spontaneous blebbing. OMV and GMMA are intrinsically very pyrogenic due to their high endotoxin content [21].

The use of adjuvants in vaccine formulations has been driven by the need to enhance vaccine immunogenicity and efficacy, especially for recombinant protein subunit antigens [22,23].

In this study, many formulations for animal testing were adjuvanted with the crystalline aluminum oxyhydroxide (AH), which is known to have a detoxifying effect due to the adsorption mechanism of the endotoxins [24]. Vaccines tested in vivo were also adjuvanted by other Adjuvant Systems (ASs). Adjuvant Systems are technologies that many pharmaceutical companies have been developing for more than two decades and are widely used in commercially available vaccines. Adjuvant Systems are based on the combination of various types of adjuvants, such as aluminum salts, oil-in-water (o/w) emulsions, liposomes, and immunomodulatory molecules, known to have an impact on the innate and/or adaptive immune responses [25,26].

To quantify the endotoxin content of the purified antigens and the corresponding drug product (DP), the Limulus Amoebocyte Lysate (LAL) test was applied [27]. This is the most widely used endotoxin detection technique prescribed by the Health Authorities and Pharmacopeias and still represents the most sensitive, accurate, and rapid method for endotoxin detection [28].

All endotoxins’ data collected in formulations used in in vivo research studies were analyzed and combined with corresponding animal welfare data reported by the internal VetCare system.

This study provides new guidelines, experimentally based, for preclinical vaccine candidates used in in vivo mice studies to set an appropriate endotoxin limit, ensuring no negative impact on animal welfare.

## 2. Materials and Methods

**Data**: Data were collected from 74 in vivo studies run in mouse models at the GSK Siena site over three years (2018–2021). These included data from 536 formulations injected into 536 groups of mice. Study size varied and depended on the single study scope, as did the statistical analysis of each individual study at the time of the execution. In detail, 436 groups, for a total of 4313 mice, were injected with *not potentially intrinsic pyrogenic* formulations, and 100 groups, for a total of 902 mice, were injected with *potentially intrinsic pyrogenic* formulations, covering a wide range of EC (from values lower than 10 EU/mL to value higher than 100,000 EU/mL). For each single animal, the following information was collected: the injected formulation, if the formulation was *not potentially intrinsically pyrogenic* or *potentially intrinsically pyrogenic*, the endotoxin content measured before each immunization, and if a VetCare report potentially correlated to the presence of endotoxins was issued, regardless the severity of the presentation.

**Formulation**: The formulations used for the in vivo studies and part of the dataset involve multiple antigens of viral or bacterial origin in different concentrations according to the different projects’ requirements. The adjuvants used in the formulations evaluated in this statistical study included not only the two main types of aluminum adjuvants, aluminum hydroxide (AH) and aluminum phosphate (AP), but also AS01 liposome-based vaccine adjuvant containing two immunostimulants, the MPL (3-Odesacyl-4′monophosphoryl lipid A) and the saponin QS-21 [29], the AS03 an oil-in-water emulsions (composed by squalene, the immunostimulant alfa-tocopherol and polysorbate 80) [30], AS04 consisting of MPL adsorbed to alum [31], and AS37 based on a synthetic Toll-like receptor 7 TLR7 agonist, which has been adsorbed to alum [32,33,34].

All formulations for in vivo testing were routinely prepared in bioburden-controlled conditions by formulation experts or by automated Hamilton Liquid Handling System using sterilized and pyrogen-free materials and solutions. Formulations were characterized prior to each immunization, and LAL test was part of the testing panel applied on all in vivo formulations. In-house-prepared formulation components were also routinely monitored for endotoxin content. Each formulation was classified as *not potentially intrinsic pyrogenic* containing purified bacterial recombinant proteins or glycoconjugates antigens with an expected low endotoxin content, or *potentially intrinsic pyrogenic*, containing OMV and GMMA antigens with an expected high endotoxin content.

**Immunization studies and animal welfare:** All animal experimentation was performed by the GSK Animal Resources Centre (ARC) in Siena (Italy) in an AAALAC-accredited animal facility, in compliance with European Directive 63/2010. All animal studies were ethically approved by the local Animal Welfare Body, as per Italian legislation. Veterinary Services supported and monitored all animal-related activities. After formulation administration, the animals’ wellbeing was monitored, and data on animal welfare were collected through an in-house system called VetCare. Through the VetCare system, any abnormal clinical presentation occurring before, during, and after administration of products was logged as animal welfare (AW) concern (detailed in the so-called VetCare reports) and assessed by ARC veterinarians. VetCare reports were classified as potentially related or not related to the effects of endotoxins according to veterinary assessments. Examples of effects not related to endotoxins are typically traumatic events (e.g., limping, fighting), stereotypical behaviors (e.g., overgrooming alopecia, barbering), or husbandry or procedural mistakes (e.g., injection mistakes). Non-specific clinical signs, such as kyphosis, labored breathing, piloerection, listlessness, and sudden death, separately or in combination, which can be caused by several causes including the toxic effects of endotoxins, were classified as potentially related to the effects of endotoxins, regardless of the severity of the clinical presentation. Immunization studies were performed in mice obtained from Charles River. In most of the studies, young adult females of the most commonly used mouse strains (i.e., CD1 and Balb-C) were used.

**Kinetic Chromogenic LAL test:** The methodological principle of chromogenic assays is to reveal the presence of the analyte in a test sample via chemically induced visible color changes. The resulting color was then measured using spectrophotometric methods to reveal the concentration of the analyte in the sample. The concentration of unknown samples was calculated from a standard curve of E. Coli-purified endotoxin. Endotoxin results were reported as UI or EU/mL. The working dilution of formulations was prepared in LAL Reagent Water (+<0.001 EU/mL) (W130 Charles River).

Charles River Endochrome-K™ reagent was used on the formulation sample and on the sample plus spike in a 96-well plate. LAL test was routinely performed by Hamilton Liquid Handling System. A maximum number of 20 sample/plate were analyzed on each analytical session. A 5-point Std Curve was freshly prepared in a range of 0.005–50 EU/mL. Plates were read on a Bioteck microplate reader. The software for endotoxin data collection, analysis, and reporting is the EndoScan-V (Version 4.3, country of origin United States of America) of Charles River.

The validity criteria of LAL test are: R-value ≥ 0.98; PPC recovery in the product is in a range 50–200% of the theoretical value; therefore, given a final concentration of PPC of a IU/mL, the acceptability range (raw IU) is from a/2 to 2a IU/mL; the EC of negative CTR (represented by LAL Reagent Water) is below the lower point of the Std Curve.

**Statistical analysis:** For each formulation, the medians of the available EC, among the results of the tests performed before each immunization, and the percentage of animals with at least one VetCare report potentially correlated to the presence of endotoxins, were computed.

Two alternative statistical methodologies, considering the EC both as quantitative variable and as ordinal one, were applied to evaluate the relationship between the EC of the formulation and the animal welfare: logistic regression [35,36] and Cochran–Armitage exact trend test [37,38].

The logistic regression was used to model the probability to observe at least one VetCare report potentially correlated to the presence of endotoxins and the EC, on logarithmic scale, of the formulation used for the immunization, using the following formula log[p(X)/(1 − p(X))] = β0 + β1Log(EC)(1) where p(X) is the probability of having observed a VetCare report, and the Log(EC) is the EC on logarithmic scale. The null hypothesis states that the coefficient β1 is equal to zero. In other words, there is no statistically significant relationship between EC and the probability of having observed a VetCare report potentially correlated to the presence of endotoxins.

The alternative hypothesis states that β1 is not equal to zero. In other words, a statistically significant relationship between EC and the probability of having observed a VetCare report potentially correlated to the presence of endotoxins cannot be excluded.

To test the null hypothesis, the overall Chi-Square value of the model was computed, and the null hypothesis was rejected for a *p*-value < 0.05 (Likelihood Ratio).

To finally evaluate the relative risk to observe a VetCare report potentially correlated to the presence of endotoxins of an increment of 10 folds in the EC, the corresponding odds ratio and its Profile Likelihood 95% confidence intervals were computed. The conclusion of no significant difference in the risk of observing a VetCare report increasing the EC by 10-fold was drawn when the confidence interval included 1. The width of the confidence interval was evaluated to exclude that the results were affected by excessively high uncertainties, which could potentially be attributed to the sample size driven by the available data.

The Cochran–Armitage trend exact test was used to test for trends in the proportions of animals with at least one VetCare report potentially correlated to the presence of endotoxins, across different levels of EC. To perform the test, two variables were created: a two-level variable indicating if the single animal had or not at least one VetCare report potentially correlated to the presence of endotoxins, and ordinal variables with three levels of EC, each defined by different cut-off points.

The binomial proportion of reported animals for each group was computed, together with the Clopper–Pearson exact 95% confidence limits [39]. The null hypothesis for the Cochran–Armitage test is no trend in the binomial proportions of animals with at least one VetCare report potentially correlated to the presence of endotoxins, increasing the levels of EC. The null hypothesis was rejected for a *p*-value < 0.05.

Finally, two thresholds were computed by consistency versus formulations that demonstrated no impact on animal welfare, one for *not potentially intrinsic pyrogenic* formulations and one for all formulations, including the *intrinsic pyrogenic* ones. As the assumption of normal distribution of the data was not verified, a non-parametric one-sided upper tolerance limit with 99% level of coverage and 90% nominal level of confidence [40] was computed to define the thresholds. The choice of a 90% nominal level of confidence ensured that the actual confidence level would not exceed 95%.

To compare the proportions of reported animals changing the EC thresholds, multiple Fisher’s Exact Tests [41,42] were performed. Fisher’s Exact Test was used to determine whether or not there is a significant association between the probability of being reported for at least one VetCare report potentially correlated to the presence of endotoxins and EC, analyzed as EC classes. The null hypothesis of independence is rejected, for the alternative hypotheses that the probability to be reported is greater for the higher EC class, when the *p*-value is lower than 0.05. Moreover, the binomial proportion of reported animals for each group was computed, together with the Clopper–Pearson exact 95% confidence limits.

## 3. Results

An extensive study to evaluate the presence of a potential relationship between endotoxin content in formulations administered to mice (the most common species used in preclinical research studies) and their welfare was conducted. Three years of historical data, from more than 500 formulations of different antigen types injected into more than 5000 mice, was evaluated.

### 3.1. Evaluation of Potential Relationship between the EC of the DP and the Animal Welfare

To investigate the potential relationship between the EC of the formulation and the animal welfare, two alternative statistical methodologies were applied: logistic regression and Cochran–Armitage trend test.

#### 3.1.1. Logistic Regression

A logistic regression modelling the percentage of animals with at least one VetCare report potentially correlated to the presence of endotoxins versus the median EC of the formulation, on the logarithmic scale, was performed. In Figure 1, a graphical representation of the investigated relation is shown, with the percentage of reported animals on the y axis and the median EC of the formulation (DP) on the x axis. An increase in the percentage of animals with at least one report is not apparent from the graph.

The overall Chi-Square *p*-value of the logistic model was higher than 0.05 (equal to 0.7357), indicating that there is no significant variation in the probability of observing a VetCare report with increasing EC.

The odds ratio for a 10-fold increase in EC was extremely close to 1 (specifically, 1.027), and its 95% confidence interval (0.875–1.187), which includes 1, confirmed a non-significant increase in the relative risk of observing a VetCare report potentially correlated to the presence of endotoxins with increasing EC. Furthermore, the upper confidence limit, being equal to 1.187, ensured a suitable precision of the estimates.

#### 3.1.2. Cochran–Armitage Trend Exact Test

A two-level variable indicating whether the individual animal had at least one VetCare report potentially correlated to the presence of endotoxins and an ordinal variable with three levels of EC were created.

The cut-off points used to define the EC levels were created based on the quartiles of the EC distribution: Group 1, minor or equal to the first quartile = second quartile (10 EU/mL); Group 2, between the second quartile and the third quartile (25.5 EU/mL); Group 3, higher than the third quartile.

No trend was apparent in the percentage of animals with at least one VetCare report potentially correlated to the presence of endotoxins in the different EC level groups (Table 1 and Figure 2), and all 95% upper confidence limit of the percentages are lower than 5%, with a maximum difference in the estimates of the percentage of animals of just 0.21% between the groups with lowest and highest EC.

The null hypothesis that there is no trend in the binomial proportions of animals with at least one VetCare report potentially correlated to the presence of endotoxins across the levels of EC was not rejected (*p*-value = 0.4251).

#### 3.1.3. Summary Results for the Evaluation of Potential Relationship between EC of the DP and the Animal Welfare

Two alternative statistical methodologies were applied to evaluate the relationship between the EC of the formulation and the animal welfare: logistic regression and Cochran–Armitage exact trend test.

Neither of the methods showed significant variation in the probability to observe VetCare reports potentially correlated to the presence of endotoxins increasing EC, supporting the hypothesis to exclude any relationship between EC of the DP and the animal welfare within the range of EC analyzed.

### 3.2. EC Threshold Definition Based on Animal Welfare

The results of the two statistical analyses reported above did not identify a significant relationship between the measured EC level and the animal welfare. The result was observed including in the analysis formulations from both categories: *not potentially intrinsic pyrogenic* and *intrinsic pyrogenic*.

Consequently, to set a threshold for EC, which takes into consideration the potential risks for animal welfare, an approach of consistency versus formulations that demonstrated no impact on animal welfare was considered.

Following the departure observed from the normality distribution of the data, a non-parametric one-sided upper tolerance limit was considered. The coverage was set at 99%, with the nominal level of confidence at 90%.

The threshold computed considering just the *not potentially intrinsic pyrogenic* formulations was 201 EU/mL, conservatively approximated to 200 EU/mL. The threshold computed including also *intrinsic pyrogenic* formulations was 89,110 EU/mL (Figure 3).

To confirm that the defined EC threshold does not significantly increase the probability of having a VetCare report potentially correlated to the presence of endotoxins with respect to the commonly accepted limit of the endotoxin level reported in [1], Fisher’s Exact Tests were performed. Two classes of EC were compared: EC ≤ 20 (EC lower than the limit reported in [1]) and 20 < EC ≤ 200 (EC higher than the limit reported on [1] and lower than the proposed threshold). A *p*-value higher than 0.05 (equal to 0.4075) allows one to reject the alternative hypothesis that the probability to observe a VetCare report is higher for the class with 20 < EC ≤ 200. The percentages of animals with at least one VetCare report potentially correlated to the presence of endotoxins in the two EC classes appeared to be comparable, with a difference in the estimates equal to 0.22% (Table 2 and Figure 4).

Finally, to confirm that a higher EC threshold for *intrinsic pyrogenic* formulations does not significantly increase the probability to have a VetCare report potentially correlated to the presence of endotoxins, Fisher’s Exact Tests were performed considering two classes of EC: EC ≤ 200 and 200 < EC ≤ 89,110. A *p*-value higher than 0.05 (equal to 0.2504) allows one to reject the alternative hypothesis that the probability to observe a VetCare report is higher for the class with 200 < EC ≤ 89,110. The percentages of animals with at least one VetCare report potentially correlated to the presence of endotoxins in the two EC classes appeared to be comparable, with a difference in the estimates equal to 0.39% (Table 3 and Figure 5).

## 4. Discussion

Endotoxins are known to represent a cause of safety issues in humans and animals, leading to pyrogenicity and toxicity. Accidental inoculation of endotoxins through vaccine administration can be a cause for health concern [43]. For this reason, control procedures are in place throughout the production and pre-commercialization (e.g., quality control testing) stages of vaccine manufacturing, to assess the presence of endotoxins in intermediate and/or final products. However, some products may be intrinsically pyrogenic, with pyrogenicity due either to endotoxic pyrogens or non-endotoxic pyrogens (i.e., pyrogenic substances not of bacterial origin) [19,20,21].

Pharmaceutical companies commonly conduct animal testing on potential vaccine candidates during the discovery and preclinical phases to assess the desired biological activity of the antigens, either alone or with adjuvants [7]. Prior to animal injection, each formulation undergoes rigorous quality control checks to minimize unnecessary animal suffering and adverse effects. It is recognized that endotoxic pyrogens can negatively impact animal health, potentially causing welfare issues or interfering with the experiment. In line with the 3R Principles [6,17], it is necessary to ensure the Refinement of procedures, Reduction in the number of animals, and Replacement of in vivo models with in vitro ones, whenever possible [6]. A key aspect of Refinement is ensuring that formulations are prepared in the best way possible to guarantee animal welfare. For these reasons, punctual and extensive controls are carried out before and after product administration to animal models. These controls aim to intercept animal welfare issues or concerns and evaluate them in the context of the experimentation. Refinement can also be achieved, therefore, through improvements in non-animal activities, such as formulation preparation and characterization [44].

Endotoxin resistance varies among animals, with mice known to be resistant and to respond to endotoxic inflammation, activating different genetic pathways from humans [11,13,14]. However, the current recommended limits (i.e., maximum dose administered) for preclinical testing in animal models are stringent, do not distinguish among different animal species, and are based only on expectations for commercial products intended for human use [1]. These limits can complicate antigen and formulation preparation for preclinical research studies, necessitating additional purification steps and controls. Any improvement in the formulation preparation, in addition to ensuring a quicker, more efficient work, tailored to species requirements, reduces risks for delays that can interfere with test scheduling [45]. Delays in administration may not be accepted in authorized protocols and may impact the overall experimentation plan with the risk of canceling or rescheduling sessions. This would impact the number of animals used in studies, again impacting the 3Rs application, in particular Reduction [46].

This study investigated the potential relationship between the EC of the drug product (DP) and animal welfare, using data collected over three years of preclinical studies at the GSK Siena site. For the first time, to our knowledge, a comprehensive evaluation of acceptable endotoxin levels in mouse models, typically used for in vivo vaccine research, was conducted, from more than 500 formulations of different antigen types injected into more than 5000 mice.

Two statistical methodologies were applied to the entire dataset, logistic regression and the Cochran–Armitage trend test, analyzing the EC both as a quantitative variable and as an ordinal variable (low, medium, high). Both statistical methods used in this study supported the initial hypothesis that there is no relationship between the EC of the DP and animal welfare within the analyzed EC range.

Finally, new Limulus Amebocyte Lysate (LAL) thresholds were statistically determined for non-potentially intrinsic pyrogenic DP (200 EU/mL) and intrinsic pyrogenic formulations (89,110 EU/mL).

It is important to note that the data utilized for this research were sourced solely from one institution and a single GSK site, which may somewhat restrict the applicability of the study’s conclusions to a broader context. However, the use of data from a single institution enhances comparability due to the consistency in how formulations were prepared, characterized, administered to the animals, and how data on animal welfare were reliably collected. This study’s evaluation is based on a comprehensive dataset, gathered over three years of preclinical studies, covering a wide range of antigens and formulation characteristics.

Furthermore, the study considered only the potential impacts of endotoxin content, although the severity of symptoms was not specifically assessed at the time of the individual studies to be correlated with EC. This methodology does not undermine the final conclusions, given that the study’s objective was to evaluate the heightened risk of mouse models being affected by the endotoxin content when non-intrinsically pyrogenic and intrinsically pyrogenic formulations were administered.

Despite the aforementioned limitations, it is worth noting that the new DP thresholds suggested for preclinical studies are significantly higher than those documented in the existing literature [1]. It has been proven that these thresholds do not notably increase the risk to animal welfare for mouse models. Moreover, these thresholds align with lethal and tolerated [8,9,10,11,12] doses, as reported in the literature.

## 5. Conclusions

To our knowledge, this study is the first to investigate the relationship between endotoxin content and animal welfare, utilizing a comprehensive dataset from preclinical in vivo studies on mouse models. The aim was not to establish the maximum tolerated endotoxin dose in these models but, rather, to identify significant endotoxin thresholds (i.e., maximum dose limits) that have been shown to not adversely affect animal welfare. Interestingly, statistical analysis revealed no correlation between the endotoxin content of formulations and the welfare of the mouse models within the evaluated range. This finding is crucial as it reduces potential risks to animal welfare in research and aids in the implementation of the 3Rs (Replacement, Reduction, and Refinement), particularly Refinement in animal research. The use of a substantial dataset and two statistical methods (logistic regression and the Cochran–Armitage trend test) to analyze the data lends credibility to the findings, bolstering confidence in the results. Despite the methodological limitations (e.g., data sourced solely from one research institute), the results can offer numerous advantages to researchers. Firstly, it can streamline the preparation and characterization of formulations, potentially saving time and resources and yielding more precise and targeted results. Secondly, the introduction of two new thresholds for non-potentially intrinsic pyrogenic DP (200 EU/mL) and intrinsic pyrogenic formulations (89,110 EU/mL) provides a practical tool for researchers.

In conclusion, this study provides valuable insights that can improve the efficiency and ethical conduct of future research involving mouse models.

## Figures and Tables

**Figure 1 vaccines-12-00815-f001:**
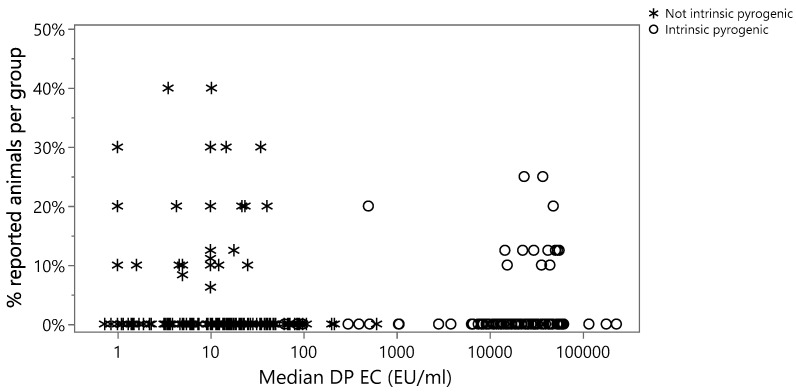
Percentage of animals with at least one VetCare report potentially correlated to the presence of endotoxins vs. median EC of the formulation (DP). Stars indicate *not potentially intrinsically pyrogenic* formulations, and dots indicate *potentially intrinsically pyrogenic* formulations.

**Figure 2 vaccines-12-00815-f002:**
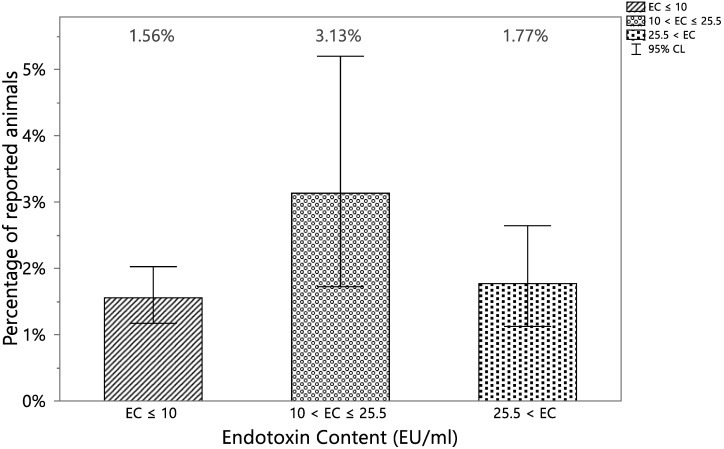
Percentage of animals with at least one VetCare report potentially correlated to the presence of endotoxins vs. different EC level groups (Group 1 EC ≤ 10 EU/mL, Group 2 10 EU/mL < EC ≤ 25.5 EU/mL and Group 3 EC > 25.5 EU/mL) with the 95% confidence intervals.

**Figure 3 vaccines-12-00815-f003:**
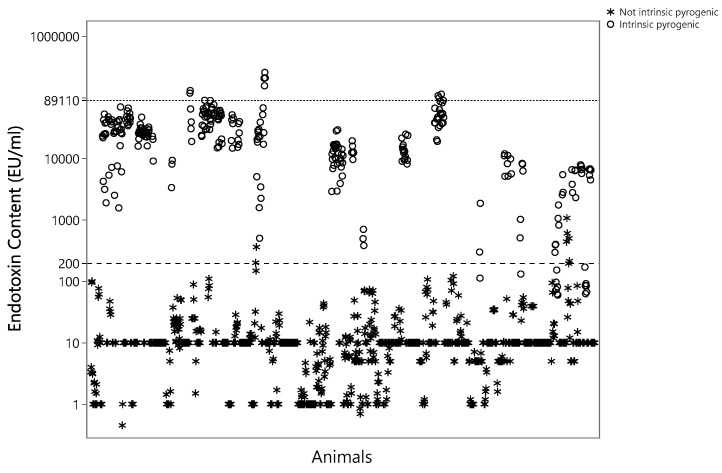
EC of the formulation (DP) injected for each animal. Stars indicate *not potentially intrinsically pyrogenic* formulations and dots indicate *potentially intrinsically pyrogenic* formulations. Dashed line corresponds to the proposed threshold for *not potentially intrinsically pyrogenic* formulations, while dotted line corresponds to the proposed threshold for *potentially intrinsically pyrogenic* formulations.

**Figure 4 vaccines-12-00815-f004:**
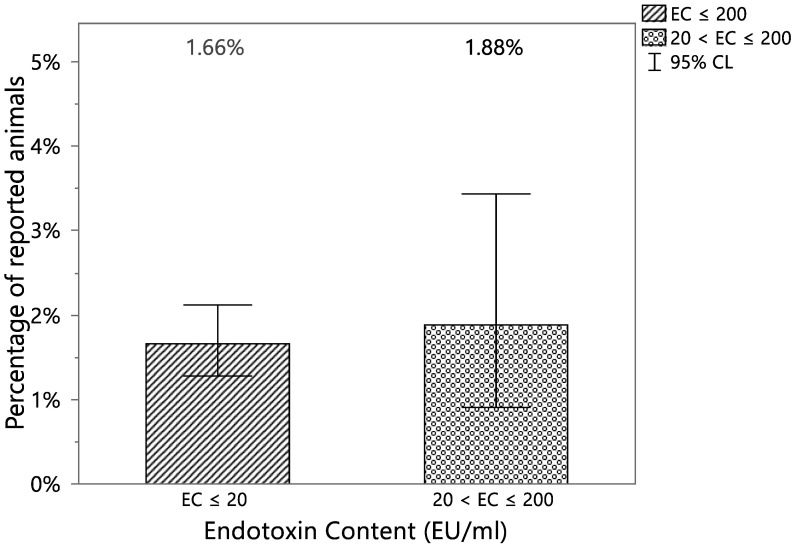
Percentage of animals with at least one VetCare report potentially correlated to the presence of endotoxins vs different EC level (classes: EC ≤ 20 EU/mL, EC lower than the limit reported in [1] and 20 EU/mL < EC ≤ 200 EU/mL, EC higher than the limit reported on [1] and lower than the proposed threshold) with the 95% confidence intervals.

**Figure 5 vaccines-12-00815-f005:**
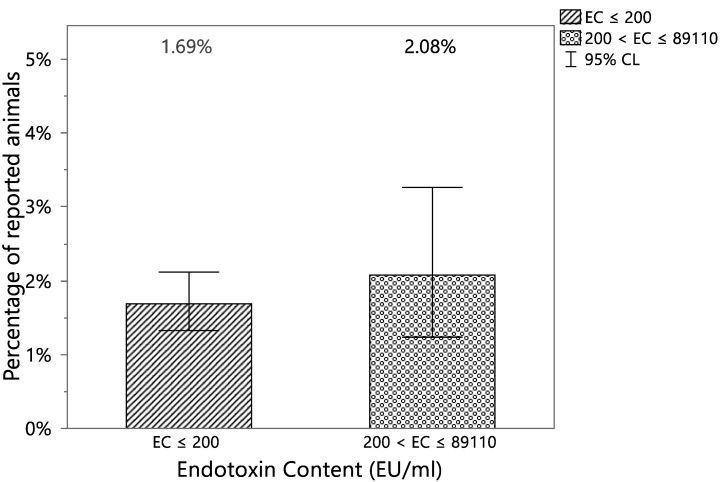
Percentage of animals with at least one VetCare report potentially correlated to the presence of endotoxins vs different EC levels EC ≤ 200 EU/mL, proposed threshold for *not potentially intrinsic pyrogenic* formulations and 200 EU/mL < EC ≤ 89,110 EU/mL, proposed threshold for *intrinsic pyrogenic*) with the 95% confidence intervals.

**Table 1 vaccines-12-00815-t001:** Contingency table of animals reported for at least one VetCare report potentially correlated to the presence of endotoxins and EC in classes created based on the quartiles of the EC distribution: Group 1 minor or equal to the first quartile = second quartile (10 EU/mL), Group 2 between the second quartile and the third quartile (25.5 EU/mL), Group 3 higher than the third quartile.

Reported	EC ≤ 10	10 < EC ≤ 25.5	25.5 < EC	Total
YES	54	14	23	91
	(1.66%)	(1.88%)	(1.88%)	
NO	3417	433	1274	5124
	(98.34%)	(98.12%)	(98.12%)	
Total	6471	447	1297	5215

EC expressed in EU/mL.

**Table 2 vaccines-12-00815-t002:** Contingency table of animals reported for at least one VetCare report potentially correlated to the presence of endotoxins and two classes of EC: EC ≤ 20 EU/mL, accepted considering the commonly accepted limit of the endotoxin level reported in [1] and 20 EU/mL < EC ≤ 200 EU/mL, higher than the limit in [1] but lower than the proposed threshold.

Reported	EC ≤ 20	20 < EC ≤ 200	Total
YES	63	10	73
	(1.66%)	(1.88%)	
NO	3729	521	4250
	(98.34%)	(98.12%)	
Total	3792	531	4323

EC expressed in EU/mL.

**Table 3 vaccines-12-00815-t003:** Contingency table of animals reported for at least one VetCare report potentially correlated to the presence of endotoxins and two classes of EC: EC ≤ 200 EU/mL, proposed threshold for *not potentially intrinsic pyrogenic* formulations and 200 EU/mL < EC ≤ 89,110 EU/mL, proposed threshold for *intrinsic pyrogenic*.

Reported	EC ≤ 200	200< EC ≤ 89,110	Total
YES	73	18	91
	(1.69%)	(2.08%)	
NO	4250	848	5098
	(98.31%)	(97.92%)	
Total	4323	866	5189

EC expressed in EU/mL.

## Data Availability

The main original contributions presented in the study are included in the article; further inquiries can be directed to the corresponding author.

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
