# Peer review of "Relationship between Endotoxin Content in Vaccine Preclinical Formulations and Animal Welfare: An Extensive Study on Historical Data to Set an Informed Threshold"

_vaccines, 2024, doi:10.3390/vaccines12070815_

Round 1

Reviewer 1 Report

Comments and Suggestions for Authors

This paper presents a study of historical data to investigate whether endotoxic content in formulations given to mice was in any way related to their welfare.

It is interesting and commendable that the authors have used historical data to answer this question, rather than performing an in vivo study.

I have 3 major points of concern that I address below, and several minor recommendations.

major points of concern:

1. Data used were from one institute. On the one hand, this allows for better data comparison, since it can be assumed that the variation in how formulations are given to the mice and data collected on welfare are more reproducible than when comparing data from different institutes or even different sites within GSK. On the other hand, this also means that the data presented here are less robust and conclusions less generic  than the authors suggest. 

2. The correlation between EC and animal welfare has been made very crude, dividing  animal welfare concerns into only 2 levels: present or not present. Although the data show that for many animals, no AW concern is present both in high and low EC studies, for the animals in which there is concern, it is not clear how badly the animals were affected. Since all data on AW concern were available, I wonder why this information has not been taken into account in the data set.

3. The discussion of the paper is not a discussion at all, but merely a copy of the abstract in different words. In a discussion, I would expect the authors to go into the deeper meaning of their findings, possible short comings of their study such as the 2 aspects I mention above and possibly recommendations for future studies. This is missing completely. 

Minor recommendations

line 45/46: the claim that mice are considered endotoxin-resistant is based on 2 references from the early 1960s, one of which is a reference concerning dogs. I believe these may be the wrong references presented here? 

Line 56/57: the authors choose an average mouse BW of 30 g. This is rather high. Later in the paper, they mention that data have been used from female mice of the BALB/c and CD-1 strain. The average weight of female CD-1 mice is indeed around 30 g, but BALB/c mice (as many other inbred strains) on average weigh no more than 20 g.

Line 163/164: Why were 2 alternative statistical methodologies applied? Are these in some way complementary to each other? 

Author Response

Response to Reviewer 1 Comments

1. Summary

2. Questions for General Evaluation

Reviewer’s Evaluation

Response and Revisions

Does the introduction provide sufficient background and include all relevant references?

Can be improved

Additional References included in the revised manuscript

Is the research design appropriate?

Yes

Are the methods adequately described?

Yes

Are the results clearly presented?

Yes

Are the conclusions supported by the results?

Must be improved

The discussion and conclusion of the paper has been completely revised by the authors, according to comments and recommendation from reviewers.

3. Point-by-point response to Comments and Suggestions for Authors

Comment 1:  Data used were from one institute. On the one hand, this allows for better data comparison, since it can be assumed that the variation in how formulations are given to the mice and data collected on welfare are more reproducible than when comparing data from different institutes or even different sites within GSK. On the other hand, this also means that the data presented here are less robust and conclusions less generic  than the authors suggest. 

Response 1: Thank you for pointing this out. We agree with this comment. Therefore, we have included considerations about the limits and the benefits of the data coming from one institute to the aim of the study in the section 4 Discussion section (lines 407-413 revised clean manuscript) and in the section 5 Conclusions (lines 442-446 revised clean version).

Comment 2: The correlation between EC and animal welfare has been made very crude, dividing animal welfare concerns into only 2 levels: present or not present. Although the data show that for many animals, no AW concern is present both in high and low EC studies, for the animals in which there is concern, it is not clear how badly the animals were affected. Since all data on AW concern were available, I wonder why this information has not been taken into account in the data set.

Response 2: Thank you for pointing this out. We agree with this comment. Therefore, we have included considerations about the methodology used to classify the animal welfare concerns, and why this methodology does not undermine the study's objective aimed at the evaluation of the increased risk of mouse models being affected by Endotoxin Content (Section 4 Discussion lines 414-419 revised clean manuscript). Data were collected at the time of the individual studies, over several years, regardless the possible correlation with EC. For the purpose of this paper we are able to take into consideration only the presence or absence of clinical presentations that could be potentially due to endotoxins, which includes therefore all cases, whether they are severely affecting animals or not.

Comment 3: The discussion of the paper is not a discussion at all, but merely a copy of the abstract in different words. In a discussion, I would expect the authors to go into the deeper meaning of their findings, possible short comings of their study such as the 2 aspects I mention above and possibly recommendations for future studies. This is missing completely. 

Response 3: Thank you for the comment, we agree with it. The section 4 Discussion has been heavily revised, also including the 2 aspects recommended by the reviewer. Additionally, section 5. Conclusions has been included in the revised version of the manuscript.

Minor recommendations

line 45/46: the claim that mice are considered endotoxin-resistant is based on 2 references from the early 1960s, one of which is a reference concerning dogs. I believe these may be the wrong references presented here? 

Response: Indeed, the references shown are not the correct ones on which we based our study. We apologise for the mistake, and we have added more appropriate references: References 7-13 in the revised clean manuscript.

Line 56/57: the authors choose an average mouse BW of 30 g. This is rather high. Later in the paper, they mention that data have been used from female mice of the BALB/c and CD-1 strain. The average weight of female CD-1 mice is indeed around 30 g, but BALB/c mice (as many other inbred strains) on average weigh no more than 20 g.

Response: Thank you for the comment. We modified the calculation considering the average weight of BALB/c mouse strain of 20 g as worst case (Section 1. Introduction lines 53-58 revised clean manuscript)

Line 163/164: Why were 2 alternative statistical methodologies applied? Are these in some way complementary to each other? 

Response: To increase the robustness of the evidence generated, two alternative statistical methodologies were applied, considering the EC both as quantitative variable and as ordinal, as added Section 2. Materials and Methods lines 173-176 (revised clean manuscript). In the Logistic Regression the EC is treated as quantitative while in the Cochran Armitage trend exact test it is treated as an ordinal variable (dividing the observations in three ordinal categories). Therefore, getting the same evidence from both the analysis increases the robustness of the results since they are based on different assumptions and on a different way to evaluate the potential link.

Reviewer 2 Report

Comments and Suggestions for Authors

I have read and reviewed with great interest the document entitled: Relationship between endotoxins content in vaccines preclinical formulations and animal welfare: an extensive study on historical data to set an informed threshold. In general, the manuscript seeks to highlight that the best-known pyrogenic impurity in vaccines is endotoxin lipopolysaccharide (LPS) from gram-negative bacteria, which can trigger inflammatory responses that can induce endotoxic shock. This fact is its main strength since the literature on endotoxic content for preclinical vaccine formulations used in animal studies is scarce and the recommended thresholds are based only on the limits of commercial vaccines established for humans, so it is not they are related to the real impact of endotoxic content on the animal welfare of species used in preclinical research studies. However, some aspects require the author’s attention, for example, the materials and methods sections, as well as the discussion, must be reevaluated and rewritten, so that readers of this paper do not have confusion when analyzing the results obtained.

For this reason, some points must be addressed to achieve publication quality. I have left some comments hoping that they can help the authors.

General comments

L44, 60, 76, 79, 88: please add a reference.

L99: Is an impact negative?

L100: please add a section indicating the methodology for capturing and revising the studies analyzed, clearly indicating the inclusion and exclusion criteria considered in your research. Likewise, add a detailed description of the experimental design of the study. Additionally, it is required to explain the sample size of each study group and what statistical method was used to determine said sample size.

L151: indicate the software version and its country of origin.

L163: Before these tests, was a statistical normality analysis performed? Please clarify.

L224: what parameters were used to evaluate animal welfare; please first clarify this aspect in the methodology section. This suggestion will make the results section clearer and more understandable for the reader.

L349: the discussion must be completely rewritten.

L350-364: The information contained in these lines must be part of the methodology, I suggest the authors move these paragraphs to that section.

L375: At the end of this section, a discussion of all limitations and perspectives identified in your study should be included. After that, it is advisable to include the conclusions.

L376: further consultation of literature related to the research work is required. I also suggest authors include more current references. Finally, the DOI identifier must be added to each reference.

Comments on the Quality of English Language

Minor editing of English language required

Author Response

Response to Reviewer 2 Comments

1. Summary

2. Questions for General Evaluation

Reviewer’s Evaluation

Response and Revisions

Does the introduction provide sufficient background and include all relevant references?

Can be improved

Additional References included in the revised manuscript

Is the research design appropriate?

Must be improved

Methodology description and study design included in the revised manuscript

Are the methods adequately described?

Must be improved

Methodology description and study design included in the revised manuscript

Are the results clearly presented?

Can be improved

Are the conclusions supported by the results?

Must be improved

The discussion and conclusion of the paper has been completely revised by the authors, according to comments and recommendation from reviewers

3. Point-by-point response to Comments and Suggestions for Authors

Comment 1:  L44, 60, 76, 79, 88: please add a reference

Response 1:  Additional references added in the revised manuscript as recommended (References 15-16; 20; 21-22; 26 revised c manuscript)

Comment 2: Is an impact negative?

Response 2: Thank you for pointing this out. We agree with this comment. Therefore, we have modified the text to clarify that it is negative impact (Section 1. Introduction line 97  revised cleanmanuscript).

Comment 3: L100: please add a section indicating the methodology for capturing and revising the studies analyzed, clearly indicating the inclusion and exclusion criteria considered in your research. Likewise, add a detailed description of the experimental design of the study. Additionally, it is required to explain the sample size of each study group and what statistical method was used to determine said sample size. 

Response 3: Thank you for the comment, we agree with it. Indeed, in the section 2 a sub-section dedicated to the detailed description of the data used for the study has been included in the revised manuscript (section 2- Materials and Methods lines 99-110 revised clean manuscript).

In response to the query about the sample size for each study group involved in our meta-analysis, and the statistical approach used to ascertain each sample size, providing this information would not add value to the discussion. This is due to the fact that the study detailed in our manuscript incorporated 74 distinct in vivo studies that were reviewed retrospectively for this paper. The sample size in each study was determined according to the unique, individual objectives of each individual in vivo study.

Comment 4: L151: indicate the software version and its country of origin.

Response 4: Thank you for the comment, we agree with it. Indeed, the information has been added in the revised manuscript (Section 2- Materials and Methods lines 163-164 revised clean manuscript)

Comment 5 L163: Before these tests, was a statistical normality analysis performed? Please clarify.

Response 5: The statistical approaches used, both to investigate the potential relationship between the EC of the formulations and the animal welfare and for setting the thresholds, do not assume the Normal distribution of the data, as for example the Student's t-test for mean comparisons, consequently an evaluation of potential deviations from Normality is not required.

Comment 6: L224: what parameters were used to evaluate animal welfare; please first clarify this aspect in the methodology section. This suggestion will make the results section clearer and more understandable for the reader.

Response 6: Thank you for the comment, we agree with it. Indeed, in the section 2 a sub-section dedicated to the detailed description of the data used for the study has been included in the manuscript ( Section 2- Material and Methods lines 106-110 and 136-150 revised clean manuscript).

Comment 7: L349 the discussion must be completely rewritten.

Response 7: Thank you for the comment, we agree with it. The section 4 Discussion has been heavily revised considering the limitations and opportunities offered by this study.

Comment 8: L350-364: The information contained in these lines must be part of the methodology, I suggest the authors move these paragraphs to that section.

Response 8: Thank you for the comment, we agree with it. The information in these lines has been moved to section 2. Material and methods (section 2. Materials and Methods lines 99-110).

Comment 9: At the end of this section, a discussion of all limitations and perspectives identified in your study should be included. After that, it is advisable to include the conclusions.

Response 9: Thank you for the comment, we agree with it. The section 4 Discussion has been heavily revised considering the limitations and opportunities offered by this study (section 4. Discussion). Additionally, a section to include the conclusions has been added to the revised manuscript (section 5. Conclusions)

Comment 10: L376: further consultation of literature related to the research work is required. I also suggest authors include more current references. Finally, the DOI identifier must be added to each reference.

Response 10: Thank you for the comment, we agree with it. The bibliography has been updated in the revised manuscript, including additional references to the research work. Finally DOI identifier has been added to the references.

Comments on the Quality of English Language Minor editing of English language required

Response: English language has been improved in the revised manuscript.

Round 2

Reviewer 1 Report

Comments and Suggestions for Authors

Review comments have been properly taken into account in the new version

Author Response

Response to Reviewer 1 Comments

1. Summary

2. Questions for General Evaluation

Reviewer’s Evaluation

Response and Revisions

Does the introduction provide sufficient background and include all relevant references?

yes

Is the research design appropriate?

Yes

Are the methods adequately described?

Yes

Are the results clearly presented?

Yes

Are the conclusions supported by the results?

3. Point-by-point response to Comments and Suggestions for Authors

Comment 1:  Review comments have been properly taken into account in the new version

Response 1: We thank the reviewer for his previous comments and suggestions that have significantly improved the quality of our manuscript.

Reviewer 2 Report

Comments and Suggestions for Authors

I thank the authors for their responses to my comments. I have observed that the manuscript has improved significantly. However, I have some final observations that I hope the authors can address, to achieve publication quality.

L42: Please elaborate on the importance of biological models in biomedical research. I suggest the authors consult and include the following reference:

  • 10.3390/ani13071223

L359, 364, 367, 370, 378, 388, 391: Please add references that support what is described.

L432: please change "the goal" to the aim. 

Author Response

Response to Reviewer 2 Comments

1. Summary

2. Questions for General Evaluation

Reviewer’s Evaluation

Response and Revisions

Does the introduction provide sufficient background and include all relevant references?

Yes

Additional References included in the revised manuscript

Is the research design appropriate?

Yes

Are the methods adequately described?

Yes

Are the results clearly presented?

Yes

Are the conclusions supported by the results?

Yes

Additional References included in the revised manuscript

3. Point-by-point response to Comments and Suggestions for Authors

Comment 1:  I thank the authors for their responses to my comments. I have observed that the manuscript has improved significantly. However, I have some final observations that I hope the authors can address, to achieve publication quality.

L42: Please elaborate on the importance of biological models in biomedical research. I suggest the authors consult and include the following reference:

  • 10.3390/ani13071223

L359, 364, 367, 370, 378, 388, 391: Please add references that support what is described.

L432: please change "the goal" to the aim. 

Response 1:  Thank you for the additional suggestions, we agree with them. Here below the changes in the manuscript we made:

-        We added in the Section 1-introduction a paragraph to elaborate the importance of biological models in biomedical research and added the suggested reference (Section 1-introduction lines 39-53 of Version 2 clear)

-        L359, 364, 367, 370, 378, 388, 391: references added (lines, 374,379,386,391,396  403,407 Version 2 clear

-        We changed the word goal with aim on L446 version 2 clear